# Ranking of Curricular Content by Pharmacy Students and Community Pharmacists

**DOI:** 10.3390/pharmacy10040071

**Published:** 2022-06-27

**Authors:** Jeffrey Taylor, Holly Mansell, Jason Perepelkin, Danielle Larocque

**Affiliations:** College of Pharmacy and Nutrition, University of Saskatchewan, Saskatoon, SK S7N 5A2, Canada; holly.mansell@usask.ca (H.M.); jason.perepelkin@usask.ca (J.P.); danielle.larocque@usask.ca (D.L.)

**Keywords:** curriculum, expectations, program evaluation, satisfaction

## Abstract

A cross-sectional survey was conducted to determine the relative rankings of 17 key components in an undergraduate program. The degree of discrepancy between curricular content and that of student and pharmacist expectations was also of interest. An online questionnaire was emailed to both groups in one Canadian province. Respondents considered four questions related to the nature and adequacy of education they did receive (pharmacists) or should receive (students) and the relative importance of key topic areas (along 11-point scales). The response rate was 31.0 (students) and 10.8 percent (pharmacists). As expected, both students and pharmacists identified therapeutics and patient counselling as critical focal points for the program, while the importance of compounding was mixed. Most topics were deemed as best handled during the didactic program, with students seeing greater value in learning a larger portion of two skills (injection training and managerial duties) post-graduation. In conclusion, discrepancies were indeed found. For students, topics such as injection training and minor ailment prescribing were perceived as receiving too little attention, while communication, pharmaceutical sciences, and professionalism received too much attention. In a significant departure in perspective, pharmacists rated communication, ethical decision-making, and professionalism almost two points higher than did students.

## 1. Introduction

In the career of academics, most will encounter students who lament the fault-line between what they feel ***should*** be the focus of their education versus what *is* actually delivered during their years of study. Students have long argued the need for more of that, less of this, and so on; it is probably therapeutic to identify such perceived shortcomings. Yet, an American survey of 11,000 graduating students in 2019 found that nearly 94 percent agreed/strongly agreed they were prepared to enter practice [1].

As students progress through a program, they will be exposed to increasing depth and sophistication, eventually receiving hands-on clinical activities at various sites. Standardized clinical examinations will often be in place to assess student performance [2]. Standards for these activities are set nationally. The Canadian accrediting body recognizes that

*pharmacy education of high quality will depend on multiple components, including general knowledge, basic and professional sciences, and professional practice experience. The pharmacy curriculum is expected to embrace the scope of contemporary practice responsibilities as well as emerging roles that ensure the rational and safe use of drugs in the individualized care of patients*.[3]

This is carried out in concert with the Association of Faculties of Pharmacy of Canada (AFPC) [4] and the National Association of Pharmacy Regulatory Authorities (NAPRA) [5]. To this end, the University of Toronto, by way of example, stated their pharmacy graduates *will deliver the enhanced scope of practice called for by the Canadian healthcare system* [6]. Steps to achieve such outcomes have been described in Canada [7,8,9]. Hall et al. outlined an approach for hospital practice experiential training in 2012 [10]. Some years later, Legal felt that the current approach to experiential education must again be re-evaluated and new approaches investigated [11]. In 2013, Slavcev et al. stated that *practicing pharmacists may not be adequately equipped to rise to the challenges associated with their expanding professional roles* [12].

Curricular content within university programs is constantly under review to ensure students are being trained as effectively as possible. However, while accreditation states what must be achieved, how it is accomplished can take many forms. Within the University of Saskatchewan, this requires (in part) feedback from students and the pharmacists they will eventually become. At times, there will be discrepancies in what faculty focus on relative to what the other two groups feel is the most relevant use of resources and curricular focal points. In this project, we examine how students and current pharmacists rank the educational focus required in 17 key areas of practice. As such, we investigate how each group (students and pharmacists) position *patient counselling* relative to *care plans*, the importance of *dispensing* in comparison to *minor ailments*, *how to vaccinate patients* versus *pharmaceutical sciences*, and so on. While a subject such as therapeutics always garners a considerable amount of attention within a program, its weighting to other topics is of interest.

## 2. Materials and Methods

### 2.1. Study Setting and Design

The design was cross-sectional in nature and utilized an online survey of upper-year pharmacy students and community pharmacists. All year 3 and 4 pharmacy students were contacted via an Advance Email (obtained from our internal mailing list) in the second week of October 2021, outlining the nature of the project and that feedback is voluntary. A link to the online site for the questionnaire (and further instructions) followed in one week. The approach for obtaining feedback from community pharmacists was via the office of the Pharmacy Association of Saskatchewan. All community pharmacists on their registry were contacted by email with the survey link. Hospital pharmacists were not included in the sample, as that group will be surveyed at a later date.

The online questionnaire used a platform from Qualtrics. Responses obtained were not tracked and therefore remained anonymous. Follow-up emails were sent to each entire group as opposed to those who were non-responders at that point (since no tracing was involved). Other than an iterative process for item wording by the authors, pilot-testing was not carried out.

Steps were taken to ensure students did not feel compelled to participate, including that feedback was anonymous and that the project was not connected to any course within the program. Given the broad nature of the questions, we anticipated that students would not feel the feedback they provided would be construed as overly negative, and thus could be freely given. Compensation was not provided to participants.

The study was reviewed by the Behavioural Research Ethics Board at the University of Saskatchewan. A full board review was deemed not necessary since it met the requirements for exemption status via program evaluation [13].

### 2.2. Questionnaire

To meet study goals, responders were asked to consider four main questions relative to the education they did receive (pharmacists) or should receive (students). The general impetus was when and how various topics should be handled. Since no document in the literature fulfilled these needs, the questionnaire was developed by the authors. The full questionnaire is presented in Appendix A. Curricular topics addressed for each main question were as follows:Care Plans, how to createCommunicating, professionalCompoundingDispensingDrug-related Problems, identifying and solvingEthical decision-makingEvidence-based medicineHospital practiceInjections, how toManagerial dutiesMinor ailment prescribingOTC therapeutics/Self-carePatient counsellingPharmaceutical sciences (kinetics, med chem, ceutics, etc.)Prescribing (emergency supply, extending supply, etc.)ProfessionalismTherapeutics

The first question posed was—*To help guide the time/resources of the PharmD program at the College of Pharmacy, what level of attention should the following topics be given?* The method for feedback was an 11-point Likert scale that spanned the width of the survey screen page. The scale was anchored from LOW (far left) to INTERMEDIATE (mid-point) to HIGH (far right). For this, responders were asked to click the numerical icon on the scale of their choosing, which best reflected their opinion for each of the 17 topics (Appendix A). For the second question, the wording was—*In a similar manner to the last question, when should the various topics be addressed for best effect?* The same scaling was provided, but with different verbal anchors at the poles and mid-point. The wording for the third question was—*Does the pharmacy program at the U of S currently devote the right amount of attention to the following topics?* Again, the scale had different anchoring. For the fourth question, the wording was—*Does the pharmacy program at the U of S currently devote enough time for practice/applying concepts covered within our list of topics?*

For each question, an option not to respond was provided if the responder felt they did not know enough about the situation. Questions related to demographics, program satisfaction, and activity level in community pharmacies were also posed.

### 2.3. Statistical Analysis

The data were stored electronically in the REDCap database and statistical analysis was performed in IBM SPSS Statistics version 28. Descriptive statistics were used to summarize the data.

## 3. Results

### 3.1. Students

The Advance Email was sent to 168 students on 4 October, the survey link on 11 October, and a reminder email on 25 October. The survey closed on 15 November. Fifty-two responded with completed questionnaires for a response rate of 31 percent.

Demographically, 78.8 percent (n = 41) were female and 63.5 percent (n = 33) were in year 4 (Table 1). Almost three-quarters indicated they had either moderate or extensive work experience in a community pharmacy. Just over 50 percent were either Neutral or Unsatisfied with their program to date.

Table 2, Table 3 and Table 4 quantify the ratings provided by students and pharmacists for the 17 topics across the four perspectives. Therapeutics and Patient Counselling rated highest for topics that required attention in the program, while Drug-related Problems and OTC Therapeutics were third and fourth, respectively (Table 2). Compounding and Managerial Duties finished at the bottom. For seven topics, students assessed they would be best handled during their didactic program, finishing below three on the scale (Table 3). Injection Training stood out as not receiving enough attention, finishing 1.6 points below the next closest topic (Table 4). Regarding enough time to practice and apply the concepts learned in the program, eight topics came in below four on the scale, indicating students could feel unprepared; how to give injections again stood out (Table 4).

### 3.2. Pharmacists

First contact was made with pharmacists on 12 October via an Advance Email to all names on the provincial registry (n = 1622). This included hospital pharmacists, who were informed a project on pharmacist education was under way and that they would be contacted at a later date. Of the emails sent, 1009 were actually opened by a recipient. Subsequently on 19 October, 1283 emails were sent solely to community pharmacists with further details of what was being asked and the link to the survey site; 710 were opened. On 29 October, 1284 reminder emails were sent (with re-mention of the link); 738 were opened. The survey closed on 15 November.

Of the 180 questionnaires attempted, 13 were removed because the pharmacist did not train at the University of Saskatchewan. Another 28 were removed due to large sections not being completed. This left 139 useable documents. Using 1283 (total emails) as the denominator leads to a response rate of 10.8 percent, whereas using 710 (opened emails) as the denominator nets a response rate of 19.6 percent.

Almost 75 percent were female (n = 102). Of the pharmacists including a graduation year (n = 137), 5 graduated in the 1970s, 11 in the eighties, 35 in the 1990s, 40 in the 2000s, 43 did so from 2010s, and 3 in the 2020s (Table 1). Most have been working full-time since graduation and staff pharmacists were just over half of the sample. Pharmacists were happier with the program they received than current students, with more than 75 percent (n = 111) Satisfied or Very satisfied.

Therapeutics, Patient Counselling, and Drug-Related Problems topped the list relative to what attention should be the focus of the curriculum (Table 2). Compounding, Hospital Practice, and Pharmaceutical Sciences rated the lowest. Between university training versus on-the-job training, Dispensing fell directly on the mid-point, suggesting that an ideal mix would include both (Table 3). As a relatively new activity for pharmacists of the province, Minor Ailment Prescribing garnered a 4.6, indicating again that an ideal mix would include both. Injection Training and Managerial Duties were assessed as receiving too little attention in the program, while Pharmaceutical Sciences were given too much (Table 4). Evidence of survey fatigue was evident at the juncture of Question 4 with pharmacists (Table 4), where numbers dropped. The means garnered for this question had less variation than what was forwarded by students, suggesting pharmacists may be more satisfied with the time allotted to practice (and apply concepts) during their program. That said, the means still leaned towards the current amount of time not being enough. Of course, for pharmacists that did not take the new PharmD program, this would be a historical afterthought.

## 4. Discussion

The first PharmD class was admitted in 2017. Development of the program has continued since that milestone. The program was designed systematically, collaboratively, and reflectively, with AFPC educational outcomes and NAPRA professional competencies as guiding documents [14]. Curriculum content was developed through extensive, collaborative processes involving faculty, staff, stakeholders, and external partners. The two guiding documents were central in all course development, as was the depth, scope, sequence, timeliness, and emphasis of the content. The curriculum was mapped to monitor educational outcomes and/or competencies. The curriculum includes foundational content, clinical practice skills, and inter- and intra-professional opportunities. The program claims *to allow students ample opportunities to practice and refine their practice skills in a variety of experiential learning settings*. Practicing pharmacists from both community and acute care settings were members of the initial working groups to ensure content and skills represented contemporary pharmacy practice. Experiential learning courses comprise a significant portion of the program, to reinforce the concepts taught, increase the complexity as the students move through the program, and allow students to advance their skills and knowledge in real-world settings.

Current students and pharmacists were surveyed to determine the degree of discrepancy, if any, between what currently is being delivered (program-wise) to that of the expectations of students and community practitioners. This brings to bear the data primarily in Questions 3 and 4, which assessed whether the right amount of attention was currently being paid such topics, and whether enough time was allotted to practice and apply concepts to relevant situations for that topic.

Discrepancies were indeed found. For students, topics such as Injection Training, Minor Ailment Prescribing, and Prescribing were perceived as receiving too little attention, while Communication, Pharmaceutical Sciences, and Professionalism receives too much attention (Table 4). Injection Training was somewhat of an outlier, finishing 1.6 points below the next closest topic. Of course, this perspective would be relative to how many hours of didactic training is currently allotted to it in the program and the year it is presented. In fact, provincial bylaws state that only licensed pharmacists can receive injection training; a bylaw change would be needed for students to receive the training.

Two topics (Dispensing and Therapeutics) were very close to the mid-point, reflecting that the current amount of attention was the right amount. No item attained a score of 7 or more, which may be reassuring to faculty. In a significant departure in perspectives, pharmacists rated Professional Communication, Ethical Decision-Making, and Professionalism almost two points higher than did students (Table 2).

Perhaps not unexpectedly, student feedback saw eight topics come in under four on the scale regarding whether they receive enough time practice and/or apply the skills needed for relevant situations (Table 4). Again, Injection Training stood out. Yet, in somewhat of a contradiction, this topic tied for third from the bottom regarding what topics the program should focus on (Question 1). Furthermore, it was near the top of subjects that students felt required more of a shift to post-graduation exposure, finishing about one point over the mid-point of the scale (Question 2).

Quite alarming was that a bigger percentage of students were Neutral or Unsatisfied with their program than those who were happy to date. COVID has had a tremendous effect on all institutions, and we were no exception; that alone may have influenced the numbers. There may also be a tendency by faculty to think that students more dissatisfied would be more apt to respond. A future goal for administration will be to determine the nature of that satisfaction, determine if it is a view held broadly by other students, and the extent that the discrepancies seen here play a role in that dissatisfaction. However, an equally valid approach could be for the reasons for such discrepancies to be better explained to students.

Seven topics were deemed by students as best handled during their didactic program, finishing below three on the scale, with another four topics under four (Table 4). Of course, neither score would suggest the absence of practical training is being implied, just that a shift in emphasis to post-graduation learning may not be ideal. Only two topics attained a mean over the mid-point—Injection Training and Managerial Duties. For those topics, students saw greater value in learning a larger portion of those skills post-graduation. That said, the results do not identify any topic that students feel would be best left fully until after graduation. These numbers were somewhat of a surprise, in that faculty often hear statements such as—*“I learned so much more from the real world after I graduated than from course work.”*

With no disregard to student input, pharmacists are likely better positioned to see the balance that should be struck between university training and skills best acquired during work experience. Managerial duties stood out as a topic leaning towards post-graduation, but even as the highest ranked topic, it still only garnered a 6.7 (Table 3). A more definitive stance, for argument’s sake, would have been a mean over 8.0, where faculty might decide to re-consider the value of the topic within the program. As seen by scores below three, strong support appeared evident for topics such as Therapeutics, Pharmaceutical Sciences, and Evidence-based Medicine to be addressed within the university program. Regarding the amount of time committed, Injection Training and Managerial Duties were assessed as receiving too little attention in the program, while Pharmaceutical Sciences was given too much. Yet, in a similar manner as seen with students, how to give injections finished only 12th of 17 topics vis-a-vis the attention they should be given in the program. A key factor in attempting to understand these data is that giving injections is a relatively new duty of pharmacists.

Not surprising was that both students and pharmacists identified Therapeutics and Patient counselling as critical focal points for the program. Training to identify and solve Drug-related problems was also given high importance. A somewhat controversial topic might be Compounding, considered by some to be a dying art not in need of much attention in pharmacy schools. Others will undoubtedly feel it is foundational to any pharmacist. At our institution, it is still an important component of the dispensing practice labs. Isolated here as a separate topic for comparison purposes, both groups gave it a similar rating regarding the attention it should receive, both being on the low end:
Q1: How much attention should it be given 
Students 4.1in the program?
Pharmacists 4.4Q2: When should it be addressed—within the
Students 4.9programor post-graduation?
Pharmacists 3.9Q3: Does it currently receive the right amount
Students 4.2of attention in the program?
Pharmacists 5.6Q4: Does the program allow enough time to 
Students 3.5practice/apply the concepts?
Pharmacists 5.4

Where there was more separation in perspectives was that pharmacists saw somewhat greater value in covering it within the program, while students saw a need for more of a balance. Pharmacists also felt it currently receives a bit more attention than what is actually needed.

Other reports, of course, have sought similar feedback. Not many, however, focused so intently on individual topics as seen with our approach. In the AACP survey of 2019, a total of 14,705 graduating students were invited to participate, with 75.1% doing so (n = 11,047) [1]. Nearly 94% either agreed/strongly agreed they were prepared to enter pharmacy practice subsequent to the education they received. The vast majority felt they would be able to apply knowledge from the clinical, foundational pharmaceutical, and biomedical sciences to the provision of patient care. Almost all (96.7%) felt they would be able to effectively communicate (verbal, non-verbal, written) when interacting with individuals, groups, or organizations. A similar percentage agree/strongly agreed they had developed the skills needed to prepare themselves for continuous professional development and self-directed life-long learning.

In Sweden, pharmacy graduates of one university were very satisfied with their program and largely felt the knowledge and skills acquired were useful in their present position (with most in community pharmacy) [15]. A majority felt their opinions as students were considered important to the faculty. Respondents were asked whether the perceptions of their education had changed over the years (a few years post-graduation compared to immediately after graduation). About half stated they had not changed their perception in the years since. A third stated they were more positive (slightly to very much) than at graduation, while 14 percent now were more negative (slightly to very much). Responders felt that lectures, seminars, and problem-based learning have been particularly useful for their career, while lab experiments and role playing were considered less useful.

Several have focused on experiential learning within a program, which ours did not (although Q3 did look at the mix of didactic versus post-graduation learning). Jacob and Boyter surveyed experiential learning coordinators in the United Kingdome, to find that universities face challenges in finding placement sites, and a need for more standardization in training [16]. Singh et al. surveyed 54 community pharmacy employers connected with the University of Wolverhampton experiential learning program [17]. The authors felt the feedback underpinned the need to examine current gaps in the pharmacy curriculum and why the changes would be needed. When pharmacists were asked what jobs students could do during placements, they described activities typical of community pharmacy workload, such as shelf-filling, stock management, and prescription reception. Providing advice on OTC medications was assessed with some confidence, while that for advice on prescription medicines and medication reviews was lower.

Skrabal et al. conducted an online survey of 4396 experiential sites within a cross-section of schools across the USA, with 1163 preceptors responding [18]. Nearly all preceptors felt that as time with a student increased, the quality of the experience rose; 20 percent felt they did not have enough time. At the time of writing, at least 30 percent of the professional curriculum was comprised of experiential education.

Margolis et al. presented data from 224 students from two graduating classes at the University of Wisconsin-Madison [19]. Pharmacy student confidence in providing evidence-based answers to clinical questions was assessed at four points in their program, with the final contact prior to graduation. The survey included five questions on skill assessment (for example, searching databases for high quality primary literature) and nine involving confidence (for example, interpreting number-needed-to-treat). Self-assessed skill and confidence improved significantly from baseline to graduation.

Katoue and Schwinghammer reviewed the literature on competency-based education in pharmacy [20]. They found that transforming a traditional pharmacy program into one that is competency-based offers benefits, but it can also be associated with significant challenges. A successful program will require significant attention to curricular design, implementation, and management.

For the current survey, sub-dividing program content into specific areas perhaps gave better insight into some curricular matters and for that reason, this approach may be of use to other institutions. That said, the depth we asked responders to engage in led to responder fatigue.

## 5. Limitations

In choosing the survey design and the selected topics, the authors were aware of the limitations of condensing program complexity into single-term entities. For example, any given responder could have an opinion that more training in diabetes is warranted, but that less is needed on musculoskeletal issues. Yet they were only able to reply from the context of “therapeutics”. The need for training on pharmacist prescribing for contraception [21] or the confidence students have in providing evidence-based answers to clinical questions [19] (by way of other examples) was left off the table. The same could be said for pharmaceutical sciences, where pharmaceutics might be deemed by some as more relevant than medicinal chemistry. Our intentions were to examine each topic relative to the others. In other words, where does managerial duties or evidence-based medicine fare in comparison to therapeutics?

The limitations of survey research, including distilling a construct such as program satisfaction down to five options, will be in play.

Many of the responders would have gone through a different program (Bachelor level) than the current PharmD, where there is more experiential training. Given that the context provided was for the current program, this could have added levels of bias. That said, preceptors of current students may be more knowledgeable with the program than non-preceptors.

The response rate for both groups was not high and thus generalizations to both populations must be performed with caution.

## 6. Conclusions

Curriculum reflection is an ongoing process, and these data will be added to other information received on a yearly basis. However, with the discrepancies seen, any action taken must be contextualized to what is currently provided in the program. For example, injection training stood out as an issue, but if more material is added, attention will have to be lessened in a different area.

Although academics spend entire careers trying to understand the educational process and then gauge the success in achieving outcomes, it would be unwise to conclude that academics know better and thus give less credence to this input. At a minimum, it would seem prudent to better support more explanation as to why we do things the way we do. Levels of frustration that are in play with current students will need to be address. That may include actual changes to the program. It could also involve reiterating the importance to students of topics such as Professional Communication.

Program evaluation is a very institutional-centric endeavor; it serves the needs of one program with likely little value to others. In this case of this work, however, the specificity inherent in the four key questions may offer new perspective to different schools in how to garner feedback on curricular content.

## Figures and Tables

**Table 1 pharmacy-10-00071-t001:** Participant Demographics.

Characteristics	Pharmacy Students		Pharmacists	
	Count	%	Count	%
	52	100	139	100
**Gender**				
Male	11	21.2	31	22.3
Female	41	78.8	102	73.4
did not answer			6	4.3
**Year of program**				
Year 3	19	36.5	-	-
Year 4	33	63.5	-	-
**Graduation year**				
1970s	-	-	5	3.6
1980s	-	-	11	7.9
1990s	-	-	35	25.2
2000s	-	-	40	28.8
2010s	-	-	43	31
2020s	-	-	3	2.2
did not answer	-	-	2	1.4
**Experience/Activity * in community pharmacy**				
None	2	3.8	1	0.7
Minimal	5	9.6	2	1.4
Somewhat	8	15.4	3	2.2
Moderate	21	40.4	33	23.7
Extensive	16	30.8	100	72
**Role since graduation**				
Staff pharmacist	-	-	78	56.1
Manager	-	-	39	28.1
Owner	-	-	20	14.4
Other	-	-	2	1.4
**Satisfaction with pharmacy education**				
Very unsatisfied	0	0	3	2.1
Unsatisfied	12	23.1	4	2.9
Neutral	17	32.7	21	15.1
Satisfied	20	38.4	70	50.4
Very satisfied	3	5.8	41	29.5

* For community pharmacists, the following qualifiers were added ‘Somewhat’ = half-time since graduation; ‘Extensive’ full-time since graduation.

**Table 2 pharmacy-10-00071-t002:** Question 1—Level of Attention Needed in the Program.

Topic	Pharmacy Students n Mean (std dev)		Pharmacists n Mean (std dev)	
Care Plans (how to create)	49	6.6 (2.1)	139	6.5 (2.1)
Communicating (professional)	49	6.7 (2.4)	139	8.6 (1.7)
Compounding	49	4.1 (2.3)	139	4.4 (2.2)
Dispensing	48	6.5 (2.4)	138	7.8 (2.0)
Drug-related Problems (identifying and solving)	49	8.7 (1.4)	139	9.0 (1.3)
Ethical decision-making	49	6.4 (2.2)	139	8.3 (1.8)
Evidence-based medicine	49	7.0 (2.2)	138	8.4 (1.6)
Hospital practice	49	6.6 (1.9)	131	5.2 (2.0)
Injections (how to)	48	5.7 (2.7)	137	7.2 (2.6)
Managerial duties	49	4.6 (2.2)	138	5.7 (2.1)
Minor ailment prescribing	49	7.4 (2.1)	138	8.1 (1.9)
OTC therapeutics/Self-care	49	8.0 (1.8)	137	8.3 (1.7)
Patient counselling	49	9.1 (1.4)	137	9.0 (1.4)
Pharmaceutical sciences (kinetics, med chem, ceutics, etc.)	49	6.2 (2.3)	136	5.5 (2.2)
Prescribing (emergency supply, extending supply, etc.)	49	7.0 (2.5)	138	7.3 (2.4)
Professionalism	48	5.7 (2.9)	137	8.3 (2.0)
Therapeutics	49	9.7 (0.7)	137	9.0 (1.5)

11-point Likert scale: Low (0), Intermediate (5), High (10).

**Table 3 pharmacy-10-00071-t003:** Question 2—Best Time to Address the Topic.

Topic	Pharmacy Students n Mean (std dev)		Pharmacists n Mean (std dev)	
Care Plans (how to create)	35	2.5 (2.3)	120	3.3 (2.4)
Communicating (professional)	40	3.9 (2.4)	126	4.5 (2.5)
Compounding	44	4.9 (2.6)	124	3.9 (2.6)
Dispensing	40	3.6 (2.3)	125	5.0 (2.3)
Drug-related Problems (identifying and solving)	37	2.2 (2.2)	114	3.8 (2.8)
Ethical decision-making	43	4.0 (2.2)	126	4.4 (2.4)
Evidence-based medicine	38	2.9 (2.6)	111	2.9 (2.7)
Hospital practice	40	4.5 (2.4)	125	5.0 (2.6)
Injections (how to)	43	5.9 (2.9)	126	4.7 (2.8)
Managerial duties	45	6.6 (2.7)	134	6.7 (2.3)
Minor ailment prescribing	41	3.5 (2.3)	121	4.6 (2.5)
OTC therapeutics/Self-care	36	2.2 (2.3)	114	3.7 (2.6)
Patient counselling	36	2.4 (2.0)	118	4.3 (2.5)
Pharmaceutical sciences (kinetics, med chem, ceutics, etc.)	35	1.8 (2.4)	96	2.1 (2.7)
Prescribing (emergency supply, extending supply, etc.)	39	4.7 (2.8)	125	5.5 (2.5)
Professionalism	42	4.2 (3.0)	119	4.7 (2.2)
Therapeutics	32	1.7 (2.3)	105	2.5 (2.8)

11-point Likert scale: During College (0), Equal Mix of Both (5), After Graduation (10).

**Table 4 pharmacy-10-00071-t004:** Questions 3 and 4—Satisfaction with Amount of Attention in the Program and Time to Apply Concepts.

Topic	Amount of Attention in Program				Time to Apply Concepts			
	Pharmacy Students n Mean (std dev)		Pharmacists n Mean (std dev)		Pharmacy Students n Mean (std dev)		Pharmacists n Mean (std dev)	
Care Plans (how to create)	43	5.9 (2.5)	117	5.7 (2.2)	40	5.4 (2.7)	100	5.4 (1.9)
Communicating (professional)	43	6.2 (2.2)	116	4.2 (1.6)	41	6.0 (2.0)	99	4.4 (1.5)
Compounding	41	4.2 (3.0)	116	5.6 (2.5)	38	3.5 (2.9)	99	5.4 (2.4)
Dispensing	41	4.9 (2.2)	113	4.1 (1.6)	39	5.2 (2.0)	97	4.3 (1.7)
Drug-related Problems (identifying and solving)	40	3.9 (1.6)	116	4.5 (1.4)	38	3.6 (1.7)	99	4.3 (1.4)
Ethical decision-making	42	4.3 (2.1)	115	4.4 (1.7)	38	3.5 (1.7)	97	4.0 (1.7)
Evidence-based medicine	42	4.7 (1.9)	116	4.7 (1.3)	40	4.3 (2.4)	99	4.7 (1.2)
Hospital practice	40	4.0 (2.0)	106	5.6 (2.0)	37	3.7 (1.7)	90	5.2 (2.2)
Injections (how to)	35	2.3 (2.9)	96	3.6 (2.1)	30	1.7 (2.6)	80	3.5 (2.1)
Managerial duties	42	5.4 (2.4)	104	3.9 (2.4)	39	3.0 (2.6)	93	3.8 (2.4)
Minor ailment prescribing	40	3.9 (2.0)	105	4.3 (1.7)	36	3.2 (1.9)	94	4.0 (1.6)
OTC therapeutics/Self-care	41	4.2 (1.3)	112	4.7 (1.3)	38	4.8 (1.4)	100	4.5 (1.3)
Patient counselling	41	4.4 (1.6)	115	4.6 (1.3)	38	4.9 (1.4)	100	4.6 (1.3)
Pharmaceutical sciences (kinetics, med chem, ceutics, etc.)	42	6.1 (1.8)	116	6.2 (2.0)	38	4.4 (2.2)	98	6.0 (2.2)
Prescribing (emergency supply, extending supply, etc.)	36	3.9 (2.5)	101	4.3 (1.5)	35	3.3 (2.4)	93	4.2 (1.6)
Professionalism	43	6.5 (1.9)	111	4.3 (1.6)	41	5.8 (2.0)	99	4.3 (1.5)
Therapeutics	40	4.8 (1.3)	114	4.7 (1.4)	36	4.4 (1.2)	99	4.5 (1.4)

11-point Likert scale: Too little (0), Right amount (5), Too much (10).

## Data Availability

Data are available upon request from the corresponding author as an SPSS file.

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
