# Peer review of "Ranking of Curricular Content by Pharmacy Students and Community Pharmacists"

_pharmacy, 2022, doi:10.3390/pharmacy10040071_

Round 1
Reviewer 1 Report
· The survey instrument seems overly casual and flawed in several ways:
o It doesn’t ask from what program (BSPharm vs. PharmD) the participant graduated.
o It doesn’t ask from what university the participant graduated, but responses are eliminated based on this.
o Although the objectives are about community practice, it does not ask the respondents to consider the questions in terms of community practice. Therefore, the perspective the respondent is supposed to take is unclear. Students may have considered all of practice while pharmacists considered only community practice.
· Because this study was only conducted with community pharmacists the discussion and conclusions should be very centered on this practice setting.
· It seems that no statistics were performed to compare the student and pharmacist responses. Thus, it is difficult to understand how the authors know there is a discrepancy between their responses.
Page 2, para 3 – Much of the audience may not understand “minor ailments” in the fashion that it is used here. Recommend a quick explanation here or in section above discussing changing roles of pharmacists. Alternatively, could use a different, more universal, term here.
Page 3, para 4 – Some of the terms in the list use jargon. Suggest not using abbreviations.
Page 12, para 3 – It is difficult to know how to interpret the low score of hospital pharmacy because no pharmacist respondents were working in the hospital.
Page 12, para 4 –Because the survey asks respondents about the current curriculum, the sentence about an historical afterthought seems unwarranted.
Page 13 – There appears to be no discussion section. Rather it seems to take place under the conclusions header.
Page 14, para 2 – The sentence beginning “There may also be a tendency by faculty . . .” seems out of place. Suggest rewording.
Page 14, para 2 – The last sentence is unclear. What move?
Page 14, para 3 – I believe the word “hear” is wanted instead of “here”.
Page 17, para 2 – The limitations of survey research, as it applies to this study should be given.
Page 17, para 3 – What type of bias could this have introduced? Was any analysis of preceptor vs. non-preceptor response done?
Page 17, para 4 – No conclusions are given in this section.
Author Response
The survey instrument seems overly casual and flawed in several ways:
I hope simple and straight-forward is not to be confused with complex and better. I have utilized the Transtheoretical Model of Change, Principal Components Analysis, etc in prior work and I feel its no better nor worse than the simple approaches taken in prior work. Flawed, okay, I will address the comments for sure.
o It doesn’t ask from what program (BSPharm vs. PharmD) the participant graduated.
We did not have to. By the date of graduation given, we knew who was BSP or PharmD. But since the response rate was so low, we did not divide responses up into PharmD or not, but rather their impressions as a whole. Also, since last year was the first PharmD graduation, we expected on a smattering of such pharmacists.
o It doesn’t ask from what university the participant graduated, but responses are eliminated based on this.
We did that during the screening process, to identify non-Sask grads. After that, all were from here.
o Although the objectives are about community practice, it does not ask the respondents to consider the questions in terms of community practice. Therefore, the perspective the respondent is supposed to take is unclear. Students may have considered all of practice while pharmacists considered only community practice.
The perspective was not community practice at all. The perspective was from community practitioners, a big difference. And in that perspective, we listed hospital practice as a field to be assessed, just like the other topics.
- Because this study was only conducted with community pharmacists the discussion and conclusions should be very centered on this practice setting.
Again, this was not the setting focus. Our focus was what community pharmacists and students felt about EVERY topic in their education. That said, we only asked comm pharms to give us that perspective, so we could see the weight they place in an area they are currently not in. We will do hosp pharms next, with the same perspective and chance to rate community pharmacy. - It seems that no statistics were performed to compare the student and pharmacist responses. Thus, it is difficult to understand how the authors know there is a discrepancy between their responses.
Notice we did not use the term "statistically significant" at any point. So, if students were 2 points higher on a scale than comm pharms, you can literally say 'they were higher'. We did not do stats due the the early stage of this work, and due to the low response rates we got. Such differences would have been easily significant via a t-test, but it is too early to get into stats at this juncture when we are still checking out the feasibility of doing more of such work.
Page 2, para 3 – Much of the audience may not understand “minor ailments” in the fashion that it is used here. Recommend a quick explanation here or in section above discussing changing roles of pharmacists. Alternatively, could use a different, more universal, term here.
Minor ailments is a universal term, but I can add explanation for sure.
Page 3, para 4 – Some of the terms in the list use jargon. Suggest not using abbreviations.
Will fix.
Page 12, para 3 – It is difficult to know how to interpret the low score of hospital pharmacy because no pharmacist respondents were working in the hospital.
That is the point here. We are doing hosp pharms next. The interesting part here was to see what eventual community pharmacists feel about getting hosp pharm content during their educational program.
Page 12, para 4 –Because the survey asks respondents about the current curriculum, the sentence about an historical afterthought seems unwarranted.
Page 13 – There appears to be no discussion section. Rather it seems to take place under the conclusions header.
This was somehow left out of the copy used for evaluation!
Page 14, para 2 – The sentence beginning “There may also be a tendency by faculty . . .” seems out of place. Suggest rewording.
Page 14, para 2 – The last sentence is unclear. What move?
Page 14, para 3 – I believe the word “hear” is wanted instead of “here”.
Page 17, para 2 – The limitations of survey research, as it applies to this study should be given.
All presented in the submission we sent.
Page 17, para 3 – What type of bias could this have introduced? Was any analysis of preceptor vs. non-preceptor response done?
Page 17, para 4 – No conclusions are given in this section.
Reviewer 2 Report
Some of what is presented as "Results" for both Students and Pharmacists in my opinion more relates to Methods; I would suggest moving text describing method and process to that section. There are two sections titled "Conclusions." I believe that much of what is included under "5. Conclusions" more relates to "4. Discussion." I suggest that the authors review this and separate "Discussion" and "Conclusions" more appropriately. The text is fine; just the category under which it is presented needs review and revision.
Author Response
I will reply to Reviewer 2 when we fix the problem at hand.
Reviewer 3 Report
Interesting study. I congratulate the authors on their innovative topic and hard work. I have a few questions.
1. As the questionnaire was not validated, would you be able to address the concerns using a non-validated instrument and how it affects interpretation of the questions by the subject using the tool and how it may affect interpretability of the results.
2. Would it perhaps have been better to separate the survey into two. One for the students and one for the pharmacists. The reason I ask is that then the questions could be worded slightly different. For example, and maybe I missed it, could the question for the pharmacists be geared to how reflect how they observed the pharmacy students in practice and how the curriculum prepared them for this. If this is implied in the questions I apologize in advance.
3. Also, I assume the pharmacists and students (APPEs only) were commenting on the curriculum as it relates to their APPEs and when they start actually working. Otherwise, how could they know what they have been exposed to is adequate if they have nothing to compare it to.
4. I don’t really understand the logic of not using the discussion section. It seems like you are “discussing” the results in the conclusion section, so why not just say this in the discussion section. Finally, you have a conclusion section and then an actual conclusion at the end. This is odd and could be resolved by populating a discussion section.
Author Response
I will address Reviewer 3 when we rectify the issue at hand.